# Study of a Quasi-Experimental Trial to Compare Two Models of Home Care for the Elderly in an Urban Primary Care Setting in Spain: Results of Intermediate Analysis

**DOI:** 10.3390/ijerph19042329

**Published:** 2022-02-17

**Authors:** Carolina Burgos Díez, Rosa Maria Sequera Requero, Jose Ferrer Costa, Francisco José Tarazona-Santabalbina, Marià Monzó Planella, Cristina Cunha-Pérez, Sebastià Josep Santaeugènia González

**Affiliations:** 1Primary Health Care Centre Passeig Maragall, Institut Català de la Salut, 08041 Barcelona, Spain; carolinaburgosdiez@gmail.com; 2Primary Health Care Centre Gran Sol., Institut Català de la Salut, 08914 Badalona, Spain; rmsequera.mn.ics@gencat.cat; 3Primary Health Care Centre Apenins, Badalona Serveis Assistencials, 08917 Badalona, Spain; jfcosta@bsa.cat; 4Geriatric Medicine Department, Hospital Universitario de la Ribera, 46600 Alzira, Spain; tarazona_frasan@gva.es; 5Medical School, Catholic University of Valencia, Sant Vicent Màrtir, 46001 Valencia, Spain; 6Centro de Investigación Biomédica en Red Fragilidad y Envejecimiento Saludable (CIBERFES), 28029 Madrid, Spain; 7Department of Surgery and Medical-Surgical Specialties, Faculty of Medicine, University of Barcelona, 08036 Barcelona, Spain; mmonzo@ub.edu; 8Faculty of Nursing, Catholic University of Valencia San Vicente Mártir, 46001 Valencia, Spain; cristina.cunha@ucv.es; 9Central Catalonia Chronicity Research Group (C3RG), University of Vic, Central University of Catalonia, 08500 Vic, Spain; 10Chronic Care Program, Ministry of Health, Generalitat de Catalunya, 08028 Barcelona, Spain

**Keywords:** home care models, preventive home visits, primary care, geriatric assessment

## Abstract

Functional dependence is associated with an increase in need for resources, mortality, and institutionalization. Different models of home care have been developed to improve these results, but very few studies contain relevant information. This quasi-experimental study was conducted to evaluate two models of home care (HC) in a Primary Care setting: an Integrated Model (IM) (control model) and a Functional Model (FM) (study model). Material and Methods: Two years follow-up of patients 65 years old and older from two Primary Health Care Centres (58 IM, 68 FM) was carried out, recruited between June-October 2018 in Badalona (Barcelona, Spain). Results of the mid-term evaluation are presented in this article. Health status, quality of care, and resource utilization have been evaluated through comprehensive geriatric assessment, quality of life and perception of health care scales, consumption of resources and complementary tests. Results: A significant difference was detected in the number of hospital admissions (FM/IM 0.71 (1.24)/1.35 (1.90), *p*: 0.031) in the Accident and Emergency department (FM/IM 2.01 (2.12)/3.53 (3.59), *p*: 0.006) and cumulative days of admission per year (FM/IM 5.43 (10.92)/14.69 (20.90), *p*: 0.003). Conclusions: FM offers greater continuity of care at home for the patient and reduces hospital admissions, as well as admission time, thereby saving on costs.

## 1. Introduction

For health services, it is a challenge to manage better the care of those with complex needs, the majority of whom are the elderly, as a consequence of increased life expectancy, which leads to more comorbidity, disability and dependency in the population [1,2,3].

Currently the percentage of the Spanish population aged 65 and over represents 19.6% of overall and will reach a peak of 31.4% in around 2050 [4].

In Spain, patients with multimorbidity represent 1.38% of the total population seen in Primary Care and comprise 5% of those seen in Primary Care over 65 years of age [5]. This leads to a considerable use of healthcare resources, including medical appointments, Accident and Emergency department visits, hospitalization and medication [1,2,3,4,5,6].

Multiple studies have analyzed different proposals to improve preventive home care for the elderly. There is evidence that Comprehensive Geriatric Assessment (CGA) based interventions for older patients are beneficial [7], and that multidisciplinary teams offer better quality of care and decrease acute care utilization among high-risk older people [8], decrease the number of cumulative days of admission [9,10] and facilitate continued living at home, largely by preventing the need for nursing home admission and reducing falls [11]. However, Mayo-Wilson et al. published in 2014 a systematic review and meta-analysis detecting many discrepancies in the studies reviewed on the impact that preventive home visits may have on patients with good baseline health or frailty, which could be attributed to the design of these studies, sample sizes or different definitions of the variables measured [12]. In any case, information on home care models of primary care in Spain is very scarce in these studies [13,14,15,16,17].

Currently in Spain, patients who cannot attend the Primary Health Care Centre (PHCC) are included in the Home Care (HC) Programme (ATDOM). Patients included in this programme must have been assessed by a doctor, nurse or social worker and meet at least the following criteria: not being able to move to the centre to be cared for, for reasons of health or physical condition or due to their social or environmental situation, temporarily or permanently [18]. 

The healthcare offered by this programme is carried out by primary care teams and involves health promotion and preventive activities, control of chronic and acute pathologies, treatment, and rehabilitation, with the aim of ensuring that patients achieve a good quality of life, along with their families, while maintaining the greatest possible autonomy [19].

Traditionally a patient’s home care in Spain is managed from the Basic Care Unit (BCU) composed by a general practitioner and a nurse who have cared for the patient since he/she first came into contact with the primary care team from the age of 15 years onwards; thus a BCU is responsible for the care of a group of people (around 1350 to 1550 patients), being the same team as that of the primary health care centre that takes care of the assigned patients, both in the centre and in home visits (integrated or traditional home care model, IM). There are other models of home care based on nursing care with the occasional support of the family doctor, or based on hospital health teams that travel to the community [10], or by interdisciplinary teams based on a reorganization of the Primary Care team that involves the creation of a home care team (family doctor and nurse) dedicated exclusively to the Home Care Programme and which are referred to as the Functional Model (FM), also called the Dispensaries Model. To date, no Spanish studies have been published that evaluate and compare the different aspects of health, resource consumption and perception of care received during 2 years of follow-up of this functional model (FM), compared to the traditional or integrated model (IM). The aim of this study is to compare both models in an urban area in caring for patients at home.

## 2. Materials and Methods

### 2.1. Study Design and Thical Aspects

This quasi-experimental study [20] compared the outcome of two Home Care (HC) models implemented in two primary health care centres in Badalona (Barcelona, Spain). The control group consisted of patients following the integrated HC model (IM) provided by the PHCC *Gran Sol*, and the study group consisted of patients following the new functional HC model (FM), linked to the PHCC *Apenins*. Both HC teams consist of a general practitioner and a nurse. In the integrated model, HC is given by the same healthcare team providing medical care at the primary care unit, with an average of 1500 inhabitants assigned to each team. By contrast, in the functional model, HC is given by a healthcare team specifically trained in the management of older, frail and multimorbid patients, providing only full-time preventive home visits and connecting to other, further required special provision services. Further details regarding the characteristics of each model are shown in Table 1. The rationale for choosing these two centres relates to the balanced demographic characteristics of the reference population (Table 2).

All the HC interventions performed in both programmes are based on current protocols designed following recommendations of the SEMFYC (Spanish Society of Family and Community medicine) and EUROPREV (European Network for Prevention and Health Promotion in Family Medicine and General Practice) inside the Program of Preventive Activities for Health Promotion (PAPPS) [22]. The study protocol was approved by the IDIAP Ethics Committee of the Jordi Gol Foundation (Approval code: P17/121). Patients (or their caregivers) voluntarily signed an informed consent, and all the information gathered was anonymized before conducting any analysis. All data was handled according to the Spanish Data Protection Law (LOPD) 15/1999 and the EU General Data Protection Regulation 2016/679. Considering the routine interventions defined in the study conventional risks were not expected to increase. Registered in ClinicalTrials.gov (Identifier: NCT03461315; 12 March 2018).

### 2.2. Selection Criteria

All patients aged over 65 years old and enrolled in the long-term HC programme at any of the two participating primary health care centres for at least 3 months were considered for eligibility. Patients were included irrespective of their cognitive status. Exclusion criteria included patients with a life expectancy of less than a month and patients with a score of 5 or more in the Pfeiffer’s cognitive impairment test (3–4 mild, 5–7 moderate, 8 or >severe) [23], who did not have a full-time caregiver or who had a part-time one, because a severe cognitive impairment is likely to interfere with the study procedure. Patients that were not registered as Badalona citizens were also excluded because it was assumed that they had temporary status, as well as patients included in a HC program due to their reduced mobility, in order to reduce bias when measuring patient-requested interventions, because the latter could not easily reach the primary health care centre facility.

### 2.3. Patient Recruitment

All subjects included in the HC programme at the two primary health care centres that met the selection criteria were contacted by phone and offered the chance to participate in the study. Patients willing to do so were scheduled a domiciliary visit to receive the study documents (i.e., the Patient Information Sheet and self-administered questionnaires/scales) and signed the informed consent themselves or, in case of cognitive impairment, via their full-time caregivers.

### 2.4. Study Conduct

The study started in June 2018 and ended in October 2020. On the first preventive home visit, once informed consents had been accepted, patients, or the caregivers in case of cognitive impairment (defined as subjects scoring 5 or more in the Pfeiffer’s test), were given self-administered questionnaires, such as EuroQoL (an instrument to complement other quality-of-life measures and to facilitate the collection of a common data set for reference purposes [24]), IEXPAC (Chronic Patient Experience Assessment Instrument [25]), and Zarit (Dependent Patient Caregiver Overload Assessment Instrument [26]). All were analyzed by the investigator, who assessed the patient’s frailty in situ. The self-administered scales were completed again by the patient and/or caregiver at the end of the second year of follow-up, during a preventive home visit. Besides these start and end visits, participants were interviewed by phone every 6 months to solve any issues and find out if any private hospitalizations or daycare centres had been used. All visits requested by either the patient or the reference doctor were also reported in a case report form (CRF). The medical professionals performing the preventive home visits were trained in the use of the scales to ensure consistency and reinforce their application.

All data, irrespective of source, were recorded in an anonymized CRF, in which patients were identified with a study code. The study investigator kept a key table with the study codes and their corresponding medical record identification codes.

In the current article the intermediate analysis of data collected in the first year of follow-up (October 2018 to September 2019 inclusive) is presented.

### 2.5. Endpoints and Variables

The primary endpoint was the difference in mean days of hospital stay per year between patients included in the integrated and functional HC programs. Secondary endpoints included the assessment of the differences between the two HC models i.e., mortality and hospital admissions, based on the IHI Triple Aim (Better Care, Better Health, Lower Costs) [27]. To this end, variables regarding subjects’ health status, quality of care, and resource utilization of patients included in the two models were compared (Table 3). The demographic characteristics of the study participants were also recorded.

Particularly, the baseline health status of study subjects included the Gerontôpole frailty screening tool and the Adjusted Morbidity Groups (AMG) risk assessment tool, which considers the type of disease, number of systems affected, and complexity of each [21,28]. Additionally, a complete baseline Comprehensive Geriatric Assessment (CGA) was performed, including the following assessments: the ability to perform normal daily tasks (Barthel scale: <20 total dependence, 20–35 severe dependence, 40–55 moderate dependence, 60 mild dependence, 100 autonomous) [29], mental health status (Pfeiffer test: ≤2 risk of cognitive impairment, 3–4 mild cognitive impairment, 5–7 moderate cognitive impairment, 8–10 severe cognitive impairment) [23], decubitus ulceration risk (Braden test: <12 high risk, 13–14 moderate risk, 15–16 < 75 years low risk, 15–18 > 75 years low risk) [30], social risk (TIRS: 1 positive indicator = social risk. Yes/no answers) [31], and social status (degree of dependency 0-1-2-3) [32]. (Seen in Table 3). The health-related quality of life of study participants and satisfaction of caregiver were assessed at baseline and at the final follow-up visit using the EuroQoL, IEXPAC, and questionnaires, respectively.

### 2.6. Statistical Analysis

The sample size calculation was based on an incidence of hospital admission of 40% and a reduction of 10% in the study group, a 2-year follow-up, and a 1:1 ratio for control and intervention groups. Under these constrains, fixed alpha and beta errors of 5%, yielded an estimated size of 581 subjects per group. The statistical power of this sample assumed alpha and beta errors of 5% was 85.3%. An intermediate analysis was performed when more than 100 subjects were recruited, with a calculated power for the sample size, comprising 126 subjects of 35.5%.

All collected variables are described for the overall study sample and for both study groups. Quantitative variables were described as the mean and standard deviation (SD), and as the median and interquartile range (IR) for normally and non-normally distributed variables, as confirmed by the Kolmogorov-Smirnov test. Categorical variables were described as frequencies and percentages. Measures of central tendency were compared using the *t*-test for independent samples or ANOVA, or their non-parametric counterparts, the Mann-Whitney U and Kruskal-Wallis tests. Categorical variables were compared using the Chi-square test or Fisher’s exact test. Post hoc analyses were performed using the Bonferroni or the Games-Howell adjustments. Variables with differences in the bivariate analysis at baseline (*p* < 0.1) and those considered clinically relevant for the authors were included in a linear multiple logistic regression to build a multivariate model in order to predict the difference in mean days of hospital stay and costs of patients in the HC program. To address Better Care and Better Health endpoints, the authors applied a binary logistic regression for mortality and hospital admission variables; age, gender, and comorbidity were included as adjustment variables. A backward stepwise regression was used to avoid overfitting of the model obtained.

A significance threshold was set at two-sided alpha value <0.05. The analysis was performed with SPSS (IBM Corp. Released 2012. IBM SPSS Statistics for Windows, Version 21.0. Armonk, NY, USA: IBM Corp.).

## 3. Results

Of the 354 patients admitted to the HC programme (113 PHCC *Apenins* and 241 PHCC *Gran Sol*) at the beginning of the study, 171 (48%) were rejected due to non-compliance with the inclusion criteria [20]. All 183 patients potentially eligible for the study were asked for consent, and 57 of them refused to participate in the study (18 PHCC *Apenins* and 39 from PHCC *Gran Sol*). A total of 126 patients, 58 (76% of those eligible) belonging to the Primary Health Care Centre with integrated home care model (PHCC *Gran Sol*) and 68 (63.5% of those eligible) attending under the functional model (PHCC *Apenins*) finally confirmed their participation in the study (see Figure 1, Follow-up chart). 

Overall, the sample of 126 patients consisted of 25% men and 75% women, and the average age of the total participants was 86.95 (±7.22) years. Both groups were comparable regarding their basic characteristics as can be seen in Table 3. There were no differences between the two populations in terms of the typology of patients in the ATDOM (home care) programme included in the study: non-complex patients (FM 5.9 vs. IM 10.3%), complex chronic patients (CCP) (FM 80.9%, IM 82.8%), and patients with chronic advanced disease (MACA) (FM: 13.2%, IM: 6.9%), nor in the implementation for these patients of an Individualized and Shared Intervention Plan (ICIP) (FM: 66.2% IM: 62.1%, *p*: 0.632), including advance directives (FM: 57.4%, IM: 55.2%, *p*: 0.806).

Although the two Primary Health Care Centers were chosen due to the similar overall socio-demographic profile of the populations attended (as shown in Table 2), it was detected that the subgroup of the population within the ATDOM (HC) programme presented greater comorbidity in those attending under the FM than in those attending under the IM, to a statistically significant degree: Adjusted Morbidity Groups (AMG) 4.5 N (%):, FM 62 (91.2), IM 43 (74.1), showing a *p*: 0.011. Significant differences were also observed in the degree of functional dependency of the population in both groups (FM/IM% grades 0.1: 60.3/81; grades 2–3: 39.7/18.9, *p*: 0.011).

FM patients were found to have more private (non-family) caregivers than IM patients (FM/IM 22 (32.4)/2 (3.4), *p*: 0.000).

Regarding the self-assessment conducted at the beginning of the study of patients in both groups, no differences were found in terms of the perception of health status (Euroqol) although differences were found in terms of the patients’ perceived experience of care (IEXPAC) (Table 3).

At the time of admission to the study, patients received a Comprehensive Geriatric Assessment (CGA) and no differences were observed in the number of patient dimensions assessed (means FM 5.05 and IM 4.36, *p*: 0.131). The results of the Comprehensive Geriatric Assessment in both groups show that both populations are totally comparable in terms of the main variables analyzed, although a more patient-centred assessment of the patient’s social needs (TIRS and Zarit) was observed in the FM than in the IM, not allowing for comparability (Table 3).

Concerning mortality at the first year of follow-up, there is no significant difference between the two models, higher in FM 20 (29.4%) compared to IM 9 (15.5%) *p*: 0.089. The multivariate analysis showed no significance differences between models for the crude OR 2.27 (95% CI 0.94–5.48; *p*: 0.069), and adjusting by age, sex, CCP, MACA and GMA there was still no statistical significance: OR 2.18 (95% CI 0.85–5.57; *p* = 0.107) (Data not in table).

There were differences in the referral of these patients to other specialists in the form of virtual consultations (13 (19.1%) FM, 4 (6.9%) IM, *p*: 0.045), but not in person (32 (47.1), 25 (43.1) FM, IM, *p*: 0.657) (Table 4).

Patients seen under the FM requested significantly more complementary tests to study their health status during this first year of follow-up: more electrocardiograms (FM/IM 28(41.2)/000 (000), *p*: 0.000), computerized axial tomography (CAT) scans (FM/IM 22%/3.4%, *p*: 0.021) and blood and urine tests (FM/IM 2.9 (3.9)/1.6 (1.5), *p*: 0.020). In contrast, no differences were found in the number of drugs prescribed, nor in the number of X-rays or ultrasound scans performed (Table 4).

During the first year of follow-up, significant differences were found in the number of home visits made by both the referring physician (FM/IM 6.25 (5.77)/3.98 (3.41), *p*: 0.008) and nurse (FM/IM 7.35 (9.50)/4.33 (5.47) *p*: 0.028). More non-referring physician visits were also detected in the FM than in the IM, FM 2.81 (2.55), IM 0.57 (1.65) significantly (*p*: 0.000). However, a trend towards less activation of the emergency medical service was observed in the FM population compared to the IM population, although not statistically significant (FM/IM 34 (50)/50 (86), *p*: 0.055) (Table 4).

As shown in Table 5, during the first year follow-up, patients treated under the FM had a lower rate of institutionalization than IM patients, FM/IM 3 (4.4)/15 (25.8), *p*: 0.003; however, there was a higher demand for respite care (RESPIR) in the population being cared for under the FM than in the IM (FM/IM 19 (27.9)/7 (12.1), *p*: 0.028, with no significant differences found in access to teleassistance or home health services between both models of care.

Concerning the consumption of health resources, during the first year of follow-up, the population treated under the FM showed a significantly lower number of hospital ward admissions (FM/IM 0.71 (1.24)/1.35 (1.90), *p*: 0.031), fewer A&E admissions (FM/IM 2.01 (2.12)/3.53 (3.59), *p*: 0.006) and fewer cumulative days of ward admission (FM/IM 5.43 (10.92)/14.69 (20.90), *p*: 0.003) (Table 5), as well as less need for activation of specialized palliative care support teams (PADES) (FM/IM 0.03 (0.17)/0.14 (0.34) *p*: 0.033). No differences were found in Home Hospitalization activations (FM/IM 0.01 (0.12)/0.19 (0.68) *p*: 0.060) and Intermediate Care Hospital admissions (FM/IM 0.21 (0.47)/0.12 (0.32) *p*: 0.239)

A multivariate analysis was performed using ANCOVA adjusting for age, sex and comorbidity categorized by AMG, obtaining a mean difference for cumulative days of hospital admission of 5.57 (SD 10.99) in the functional model compared to 13.88 (SD 16.91), *p* < 0.001. ANCOVA analysis was repeated including, in addition to age, sex and comorbidity, degree of dependency, private caregiver, and overburden as variables. In addition, the new results showed a mean difference for cumulative days of hospital admission of 6.04 (SD 11.38) in the functional model compared to 13.09 (SD 16.44), *p* < 0.001. (Data not in table).

## 4. Discussion

The present study shows clearly differentiated health outcomes in two populations with similar socio-demographic characteristics (except for a higher comorbidity at the time of inclusion in the FM population), treated under two different models of home care.

Despite being models of healthcare with similar characteristics in terms of patients, based on IGV and individualized care plans, the FM shows a greater intensity of home follow-up, with a greater number of visits and complementary examinations to study the health status of the assigned population. As already evidenced in Stuck’s systematic review [33], this fact has clear benefits for the health status of the population under FM. The present study found that the population under FM showed, already in the first year of follow-up (despite having a higher comorbidity than those under IM), a lower risk of institutionalization, admission to acute hospital, emergency care and a higher probability of continuing to live at home after one year despite needing respite care (minimum 30 days per year) [34]. This is possibly related to the initial situation of families cared for under FM, in which the caregiver is more overburdened (see Table 3), although it was not possible to compare this possibility in this first analysis.

The authors consider that the increase in the consumption of intermediate products in the population under FM is associated with the greater comorbidity of these patients (compared to those treated under IM), and probably also with the greater proactivity and follow-up of such patients due to the greater number of follow-up visits made during this first year. This is consistent with different published articles [35,36], to the point of being considered a predictor of healthcare expenditure, since this population, known as high need/high cost [37], is in fact the one that concentrates healthcare expenditure and has the highest risk of mortality. This is confirmed in our study, although there is no statistical significance in terms of mortality.

Recently, a study on the characteristics and resource consumption of PCCs showed the need to find efficient and evaluable models of social and health care [1]. Accordingly, our study shows that the accessibility and intensive follow-up of patients cared for at home under the FM does not lead to an increase in the number of referrals to other specialties or a greater number of pharmacological prescriptions, compared to the IM. Therefore, by being more accessible, the FM could plan the overall care of the person’s needs in a more individualized way, adapting this plan to the evolutionary characteristics (comorbidity situation, dependence, family environment, etc.), which results in greater dedication from the team, avoiding unnecessary referral to other specialists and not over-prescribing drugs (exercising a more person-centred vision of prescription, as has been reported in other research). [38].

The FM shows a better resolution of healthcare crises in its population attended at home resulting, as shown in the follow-up at year one, in a lower number of admissions to the A&E and a lower number of admissions to a hospital ward, with a clear trend towards a lower activation of the EMS (*p*: 0.055), possibly not significant due to the sample size obtained. This result is consistent with the publication of Vila et al. [10] in 2017 in which, with a population of 261 people of similar mean age, it was shown that there was a reduction in the number of cumulative days of admission per year from 3.5 to 1 day (*p* < 0.001) after including patients in a multidisciplinary care programme that included professionals from Primary Care and Hospital Care.

In this mid-term evaluation of this study, it was found that there is a highly significant difference in the number of cumulative days of admission per year between the two models of care, being lower in the FM than in the IM (FM/IM 5.43 (10.92)/14.69 (20.90), *p*: 0.003) with the repercussion on costs that this represents. In line with what was published this year comparing the institutionalized population with patients who remained at home, revealing important clinical, demographic and mortality differences [39], the present study shows that patients treated in the FM remain at home more often, despite the fact that no greater social risk was detected in the IM than in the FM.

Finally, at the time of recruitment, no differences were found in both models in terms of the perception of quality of life. However, differences were observed in terms of the experience of care received. FM implies a change in the healthcare team from the moment the patient is admitted to the Home Care Programme, leaving aside longitudinally, which is one of the main pillars of primary care and which has been classically associated with better health outcomes than decentralized care. Results will be re-evaluated at the end of the study to find out whether or not the results are consistent with those published by Hogg et al. [40], showing that patients experience an improvement in their subjective quality of life with the introduction of multidisciplinary care models, or with other research, such as that conducted by Marta Gorina et al. [14] in 2013, which concluded that populations cared for under a FM have a higher degree of satisfaction and perceived quality of care, although the tool used to assess this fact is different (IEXPAC in the present study vs. Satisfaction Assessment of Home Care Service (SATISFAD-12).

As mentioned in the calculation of the sample size, due to the characteristics of the patients treated under the primary healthcare domiciliary models, it was expected that there would be a 15% loss of sample over the course of the study. However, in this first year of follow-up, losses amounted to 42.6% in the FM and 22.4% in the IM, of which 31% and 30.7% were due to transfers of the patient to another health area and the rest to exitus, respectively.

### Limitations

A limitation of the present study is the impossibility of carrying out a randomized clinical trial due to the type of population and the services provided. Both groups presented significant differences at baseline in comorbidity, dependency, caregiver support and caregiver over-burning that could possibly influence the results obtained, constituting a selection bias. Certainly, both groups are not similar at baseline, but the differences have been minimized via the statistical approach. Although these variables were included as adjustment variables in the multivariate analysis, there is a selection bias.

As this is a comparison study of two healthcare models in two different Primary Healthcare Centres, a quasi-experimental study has been designed with a possible Hawthorne effect (participants in a study may alter their behavior when they are aware of being observed) [41], which is not avoidable.

Likewise, the sample size is reduced to the population included in the Home Care Programme, affecting patient recruitment, as it represents a small percentage of the total population. The *Gran Sol* PHCC is located in an urban area with many architectural barriers that impede patients from reaching the PHCC, and who are therefore attended at home, despite having one or no chronic pathology. To avoid selection bias, patients included in the HC programme were excluded from the study protocol. This justifies the fact that in the Gran sol PHCC there were, initially, 114 patients who did not meet the inclusion/exclusion criteria, as shown in the arrow diagram (Figure 1). Similarly, the study potential is reduced by the fact that some of the patients are MACA, or are in their last days, or in a situation of fragility and refuse to participate in the study.

As this is the frailest and most comorbid segment of the population, many mortality losses were detected.

In this study, two Primary Care Health Centres located in densely populated areas were compared. The authors question whether the results obtained would be expected in other population settings, such as suburban or rural areas. The power of the recruited sample size is low, as expected for an intermediate analysis. Despite this low power, beta error increases, resulting in differences which are more difficult to find, when they do exist. However, the results are conclusive about the advantages of the functional model over the integrated model.

## 5. Conclusions

In the present comparative study of models of preventive home visits in Primary Care, it is observed that the FM is a more accessible model, of higher effectiveness and greater efficiency in the consumption of acute hospitalization resources, in spite of attending a population with greater morbidity and mortality due to greater comorbidity based on AMG. In addition, it has a favorable impact on permanence at home throughout this first year, avoiding the institutionalization of patients who require a high level of social support at home (as can be seen in the need for respite care or the need to hire a home caregiver) which, when not guaranteed (integrated social and health care at home), leads to greater institutionalization (population under IM). The present study has been carried out in a densely populated city and it would be interesting to test the results also in semi-urban or rural settings. Likewise, due to the high mortality (higher than expected in the initial sample design in both populations), it would be advisable to apply the model to a larger population so that an analysis with greater statistical power could be carried out.

## Figures and Tables

**Figure 1 ijerph-19-02329-f001:**
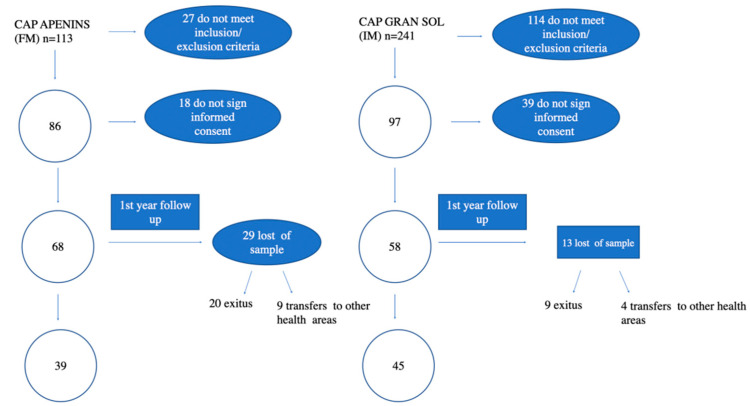
Flow diagram of the studied simple. CAP: PHCC Primary Health Care Centre. LOST OF SAMPLE: loss of patients due to death or transfers to others health areas.

**Table 1 ijerph-19-02329-t001:** Main characteristics of the two investigated models.

Characteristics of the Healthcare Team	Integrated HC	Functional HC
**Team composition**	Nurse and family physician	Nurse and family physician.
**Team function**	The same healthcare team provides HC and manages patients in the primary health care centre independently of their care needs (prevention, health promotion, patients with complex needs, patients in HC program or patients at end of life).	The healthcare team is dedicated exclusively to HC.
**Interprofessional communication**	Healthcare professionals are part of the healthcare team regularly managing patients in the primary health care centre.	Although not managing patients in the primary health care centre, the HC team is part of the health care staff of the centre and their members participate in the centre meetings as specialists
**Training**	Regular training of family doctors, including regular stays at mental health and geriatric units. Regular training of nurses.	Regular training of family doctors, including regular stays at mental health and geriatric units. Nursing staff and doctor receives additional training regarding the management of chronic patients, fragility, and palliative care. Continuous updates.
**Type of professional in each visit**	Nurse, family doctor or both.	Nurse, family doctor or both.
**Preventive visits**	Visits of nursing staff scheduled based on the monitoring requirements of each disease as established by local guidelines. Visits of physician scheduled at physician’s discretion based on the disease progression and clinical status of patients.	Visits of nursing staff scheduled based on the monitoring requirements of each disease as established by local guidelines. Visits of physician scheduled at physician’s discretion based on the disease progression and clinical status of patients.
**Dedication to the type of care activity**	90% Care at the health centre, 10% at home (depending on the organisation of the centres).	100% Home care
**Non-urgent** **acute visits**	The patient calls the centre and the physician schedules the visits at home in a deferred way, according to agenda.	During working hours, the patient directly contacts the physician of the HC team. Outside working hours: the patient calls the centre and the physician available at that moment (not always the one regularly visiting the patient at the primary health care centre) visits the patient at home.
**Urgent visits**	The patient calls the PHCC and a doctor from the centre, who is on call, sees him/her (this may not be the patient’s usual doctor).	The patient calls the HC team until 15:00. From 15:00 to 20:00, the patient calls the PHCC and a doctor from the centre, who is on call, sees him/her (this may not be the patient’s usual doctor).
**Financial approach**	All visits are fully covered by the public health system.	All visits are fully covered by the public health system.

HC: home care PHCC: Primary Health Care Centre.

**Table 2 ijerph-19-02329-t002:** Characteristics of the participating centres ^a^.

	Integrated HC (PHCC Gran Sol)	Functional HC (PHCC Apenins)	*p*
**Location**	Badalona, Catalonia, Spain	Badalona, Catalonia, Spain	
**Professional profile**	MDs and nurses specialized in family medicine	MDs and nurses specialized in family medicine	
**Reference population ^b^, No.**	19.442	19.043	
**Over-Aging index ^c^,%**	11%	9.2%	<0.001
**Foreign population ^d^, **n** (%)**	3499 (17.9%)	3046 (15.9%)	<0.001
**≥65 years old, **n** (%)**	3480 (17.9%)	2970 (15.6%)	<0.001
**AMG, adjusted indicator (IC 95%)**	1.189 (1.173–1.206)	1.178 (1.161–1.195)	–
**Mortality, annual (%)**	7	5.7	0.143
**IT application**	eCAP	eCAP	

HC: Home Care, PC Primary Care, MD Medical Doctor, AMG Adjusted Morbidity Groups [21]; IT Information Technology. **^a^** Differences between PHCC *Gran Sol* and PHCC *Apenins*. **^b^** Data from Msiq (Generalitat de Catalunya©), period between January and December 2015. **^c^** The number of persons aged 74 or over per total of persons over 64 years old. **^d^** The number of subjects with a foreign nationality.

**Table 3 ijerph-19-02329-t003:** Socio-demographic variables and baseline CGA ^1^ outcomes.

	Apenins	Gran Sol	*p*
	(Functional Model)	(Integrated Model)	
	*n* = 68	*n* = 58	
Average age	86.66 (7.6)	87.2 (6.7)	0.39
Age %:			0.457
Group 1 (between 65 and 74)	5.9	3.4	
Group 2 (between 75 and 84)	33.8	25.9	
Group 3 (>=85 years)	60.3	70.7	
Sex: (%)			
Male	23.5	27.6	0.602
Female	76.5	72.4	
Typologies of patients in the programme ATDOM ^2^ (%):			
Patients with non-complex medical problems	5.9	10.3	0.365
Chronically complex patient (CCP ^3^)	80.9	82.8	
Chronically ill patients with advanced disease (MACA ^4^)	13.2	6.9	
ICIP ^5^ realizado n (%)	45 (66.2)	36 (62.1)	0.632
ICIP with PDA ^6^ n (%)	39 (57.4)	32 (55.2)	0.806
Adjusted Morbidity Groups (AMG ^7^) n (%)			0.011
Group 1 (1,2,3)	6 (8.8)	15 (25.9)	
Group 2 (4,5)	62 (91.2)	43 (74.1)	
Degree of dependency (average)			0.011
0–1	41 (60.3)	47 (81)	
2–3	27 (39.7)	11 (18.9)	
TIRS ^8^ n (%)	6 (11.5)	12 (26.6)	0.056
No falls n (%)	63 (92.6)	53 (91.3)	0.957
No presence of decubitus ulcers n (%)	62 (91)	52 (89)	0.475
Barthel	55.15 (25,8)	60.5 (21,4)	0.262
Pfeiffer	3.94 (3.2)	2.83 (3.0)	0.078
Braden	17.75 (2.6)	17.64 (2.4)	0.824
Private caregiver No. (%)	22 (32.4)	2 (3.4)	0
Euroqol (subjective assessment)	4.75 (2.32)	4.35 (1.87)	0.291
IEXPAC ^9^	5.85 (1.69)	5.98 (1.17)	0.004
Caregiver overburden (Zarit)	58.08 (17.1)	29.27 (27.8)	0.001

^1^ CGA: Comprehensive Geriatric Assessment. ^2^ ATDOM: Home Care Programme. ^3^ CCP: Chronically Complex Patient. ^4^ MACA: Chronically Patients with Advanced Disease. ^5^ ICIP: Individual and Shared Intervention Plan. ^6^ PDA: Advance Healthcare Directive Plan. ^7^ AMG: Adjusted Morbidity Groups. ^8^ TIRS: Social Risk Indicator Scale. ^9^ IEXPAC: Chronic Patient Experience Assessment Instrument.

**Table 4 ijerph-19-02329-t004:** Health care needs during the first year of follow-up.

	Apenins (Functional Model) *n* = 68	Gran Sol (Integrated Model) *n* = 58	*p*
Online consultations with a hospital specialist No. (%patients)	13 (19.1)	4 (6.9)	0.045
In-person referrals to hospital specialists No. (%patients)	32 (47.1)	25 (43.1)	0.657
ECG ^1^ (%patients)	28 (41.2)	0.00 (0.00)	0.000
Conventional XR ^2^ requested No. (%patients)	44.2%	13.7%	0.053
Ultrasound scans requested No. (%patients)	16.1%	8.6%	0.445
CAT ^3^ requested No.(%patients)	22%	3.4%	0.021
Blood and urine tests (mean + STD ^4^)	2.9 (3.9)	1.6 (1.5)	0.020
Prescribed drugs (mean + STD)	10.05 (3.5)	9.81 (5.13)	0.757
Online consultations +G.P. ^5^ BCU ^6^ (mean + STD)	9.90 (6.27)	9.07 (6.74)	0.477
In-person consultations G.P. BCU (mean + STD)	6.25 (5.77)	3.98 (3.41)	0.008
In-person consultations G.P. non- BCU (mean + STD)	2.81 (2.55)	0.57 (1.65)	0.000
Online consultations NUR ^7^ BCU (mean + STD)	1.99 (3.23)	4.90 (5.16)	0.000
In-person consultations NUR BCU (mean + STD)	7.35 (9.50)	4.33 (5.47)	0.028
In-person consultations NUR non- BCU (mean + STD)	3.24 (9.35)	5.33 (10.35)	0.238
MES ^8^ activation No. (%)	34 (50)	50 (86.2)	0.055
Admissions to private nursing homes No. (%)	3 (4.4)	15 (25.8)	0.003

^1^ ECG: Electrocardiogram. ^2^ XR: X-rays. ^3^ CAT: Computerized axial Tomography. ^4^ STD: Standard Deviation. ^5^ G.P.: General Practitioner. ^6^ BCU: Basic Care Unit. ^7^ NUR: nurse. ^8^ MES: Medical Emergency system.

**Table 5 ijerph-19-02329-t005:** Health and social outcomes after one year follow-up.

	Apenins (Functional Model) *n* = 68	Gran Sol (Integrated Model) *n* = 58	*p*
Respite care (up to 30 days) (Respir ^1^)	19 (27.9)	7 (12.1)	0.028
Teleassistance	60 (88.2)	54 (93.1)	0.353
HHS ^2^	53 (77.9)	50 (86.2)	0.231
Admissions to hospital ward (No.)	0.71 (1.24)	1.35 (1.90)	0.031
A&E ^3^ admissions (No.)	2.01 (2.12)	3.53 (3.59)	0.006
Admission in Intermediate Care Hospital. (No.)	0.21 (0.47)	0.12 (0.32)	0.239
No. of cumulative days of admission (on ward) per year.	5.43 (10.92)	14.69 (20.90)	0.003
No. Admission in Hospital at home	0.01 (0.12)	0.19 (0.68)	0.060
No. Admission in PADES ^4^	0.03 (0.17)	0.14 (0.34)	0.033

^1^ RESPIR: Limited temporary stays in private residential centres for the elderly or provision of private home care services for the elderly financed by the Barcelona City Council. ^2^ HHS: Home Help Service. ^3^ A&E: Accident and Emergency Department. ^4^ PADES: palliative care support team.

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
