# Peer review of "Study of a Quasi-Experimental Trial to Compare Two Models of Home Care for the Elderly in an Urban Primary Care Setting in Spain: Results of Intermediate Analysis"

_ijerph, 2022, doi:10.3390/ijerph19042329_

Round 1
Reviewer 1 Report
This study aims to compare two models of home care using quasi-experimental trial. While this article addresses an increasingly health issue and quasi experimental design is powerful in evaluating health policies, too much emphasis has been put on the study design, instead of the implication and value of this study. also, it will be helpful to add the study setting (urban Spain) in the title.
For example, in abstract the first sentence should briefly describe why home care for elderly matters, and the differences between the two models. The “methods” session in abstract should briefly describe what kind of data has been collected and what indicators are used for assessment.
Introduction
“In Spain, patients with multimorbidity seen in primary care represent 1.38% of the 49 total population and comprise 5% of those over 65 years of age” , it is not clear what is the denominator
The introduction session is very informative, explaining the health burden and role of PHC/home care in the health service delivery system. it might be helpful to add information about what kind of patients are eligible for PHC/home care (insurance? disease conditions, etc.)
Last paragraph, I understand this is the first study of this kind in Spain, but a proper literature review should be conducted and reported on similar studies conducted in other countries, and state clearly why it matters to conduct one in Spain.
Methods
First two sentences, please correct the gramma issue. There is another sentence not in English under the subtitle “statistical analysis”. Please do a comprehensive check of incomplete sentences and gramma issues.
There are many abbreviations without proper explanations/definitions, please check.
Results
Number of ineligible patients, and patients refused to participate in the study seem to be very different among two groups. i.e., nearly half of the patients in IM group didn’t meet the eligibility criteria. Does that mean patients in this PHC might have more severe conditions, comparing to the other group. Please explain the potential selection and/or loss-follow up bias in this study, any efforts to control the bias, and the implications on findings.
Table 3, the display is weird and misleading, I guess the author should use “.” to replace the “,”.
While the demographic composition of two groups seem to be comparable, there are some noticeable difference between the two, including morbidity, degree of dependency, caregiver, etc in baseline, and there are many differences in the health care needs between two groups.
For those conditions, it is challenging to attribute the health benefit of certain group to the HC model, considering the two groups may not be comparable. There are some statistical approaches that can help. A decomposition analysis might be able to address some of the differences in baseline/health need. A propensity matching score can also be considered.
Considering the differences between two groups, the conclusion and findings of this study is relatively weak. However, it is usually a common limitation of health care study. it is not possible or too expensive to conduct RCT. Therefore, I believe this study still has it’s value in describing the baseline, health needs and follow up results of two healthcare model. I would appreciate if authors can tone down the conclusions, perform additional analysis to address some of the potential bias.
Author Response
We would like to thank both reviewers for their thorough review of the manuscript. As detailed in the specific responses to each comment, we have revised the manuscript by including the majority of the suggestions. We believe that, thanks to these suggestions, the manuscript is now clearer, especially for readers who are unfamiliar with healthcare settings.
Reviewer 1…………………………………………………………………………………………..
Comment 1: This study aims to compare two models of home care using
quasi-experimental trial. While this article addresses an increasingly
health issue and quasi experimental design is powerful in evaluating
health policies, too much emphasis has been put on the study design,
instead of the implication and value of this study. Also, it will be
helpful to add the study setting (urban Spain) in the title.
For example, in abstract the first sentence should briefly describe
why home care for elderly matters, and the differences between the two
models. The “methods” session in abstract should briefly describe what
kind of data has been collected and what indicators are used for
assessment.
Response: We appreciate the reviewer's suggestion, since it clarifies the scope of the study and provides information, right in the abstract, on the methodology followed and the impact that the results obtained could have.
Comment 2: Introduction
“In Spain, patients with multimorbidity seen in primary care represent
1.38% of the 49 total population and comprise 5% of those over 65
years of age” , it is not clear what is the denominator.
Response: We appreciate the comment and agree that the information given could be better defined, so we have modified the formulation of the paragraph, detailing what we consider to be multimorbidity and stressing the fact that it represents the population actually attended by Primary Care. In Spain, patients with multimorbidity seen in primary care represent 1.38% of the total population (denominator: population attended in Primary Care) and comprise 5% of those over 65 years of age (denominator: population attended in Primary Care older than 65 years of age).
Comment 3: The introduction session is very informative, explaining the health
burden and role of PHC/home care in the health service delivery
system. It might be helpful to add information about what kind of
patients are eligible for PHC/home care (insurance? disease
conditions, etc.)
Response: We would like to thank you for your suggestion. We have added the criteria defined in the Programme to improve home care from Primary Health Care of the Catalan Institute of Health of the Catalan Government: patients included in this programme must have been assessed by a doctor, nurse or social worker and meet at least the following criteria: Not being able to move to the center to be cared for, for reasons of health or physical condition or due to their social or environmental situation, temporarily or permanently.
Comment 4: Last paragraph, I understand this is the first study of this kind in
Spain, but a proper literature review should be conducted and reported
on similar studies conducted in other countries, and state clearly why
it matters to conduct one in Spain.
Response: According to your suggestion, we have added the reference of a related doctoral thesis, an oral communication in a Primary Care Congress, and references of studies comparing home care models with respect to some specific aspects. Some studies have been carried out in Spain comparing home care models, however, they differ in their presentation, approach and follow-up.
Methods
Comment 5: First two sentences, please correct the gramma issue. There is another
sentence not in English under the subtitle “statistical analysis”.
Please do a comprehensive check of incomplete sentences and gramma
issues.
Response: We appreciate the feedback. The text has been revised and corrected.
Comment 6: There are many abbreviations without proper explanations/definitions,
please check.
Response: We agree with the suggested revision and have corrected them.
Results
Comment 7: Number of ineligible patients, and patients refused to participate in
the study seem to be very different among two groups. i.e., nearly
half of the patients in IM group didn’t meet the eligibility criteria.
Does that mean patients in this PHC might have more severe conditions,
comparing to the other group. Please explain the potential selection
and/or loss-follow up bias in this study, any efforts to control the
bias, and the implications on findings.
Response: The Gran Sol PHCC is located in an urban area with many architectural barriers that impede patients from reaching the PHCC and are therefore attended at home despite having one or no chronic pathology. To avoid selection bias, patients included in the HC programme were excluded from the study protocol. This justifies the fact that in the Gran sol PHCC there were, initially, 114 patients who did not meet the inclusion/exclusion criteria, as shown in the arrow diagram (Figure 1). We have explained this issue in Limitations subsection.
Comment 8: Table 3, the display is weird and misleading, I guess the author
should use “.” to replace the “,”.
Response: We agree with the suggested writing, we have corrected the error.
Comment 9: While the demographic composition of two groups seem to be comparable,
there are some noticeable difference between the two, including
morbidity, degree of dependency, caregiver, etc. in baseline, and there
are many differences in the health care needs between two groups.
For those conditions, it is challenging to attribute the health
benefit of certain group to the HC model, considering the two groups
may not be comparable. There are some statistical approaches that can
help. A decomposition analysis might be able to address some of the
differences in baseline/health need. A propensity matching score can
also be considered.
Considering the differences between two groups, the conclusion and
findings of this study is relatively weak. However, it is usually a
common limitation of health care study. it is not possible or too
expensive to conduct RCT. Therefore, I believe this study still has
it’s value in describing the baseline, health needs and follow up
results of two healthcare model. I would appreciate if authors can
tone down the conclusions, perform additional analysis to address some
of the potential bias.
Response: The authors thank reviewer #1 for the comment. The authors agree with the presence, at baseline, of quite a few differences in the health care needs between two groups. This is one of the usual problems in research when it is not feasible to carry out a randomized clinical
trial. For this reason, we will emphasize in the Limitations subsection the presence of these differences and their possible implication in the results obtained. The groups certainly differ in these aspects, although it is also true that not all the differences benefit the intervention under study. Therefore, the variables with statistically significant differences were included in the multivariate analysis and the ANCOVA analysis was repeated including, in addition to age, sex and comorbidity, the variables degree of dependence, private caregiver and overburden. We find the statistical analysis proposed by reviewer #1 very interesting, though given the size of the sample, we consider that a propensity matched score or a decomposition analysis would excessively fragment the sample, losing statistical power. In this context, and with all the limitations expressed, we consider that the best approach proposed is the one commented on.

Reviewer 2 Report
Please, see the attachment.

Author Response
We would like to thank both reviewers for their thorough review of the manuscript. As detailed in the specific responses to each comment, we have revised the manuscript by including the majority of the suggestions. We believe that, thanks to these suggestions, the manuscript is now clearer, especially for readers who are unfamiliar with healthcare settings.
Reviewer 2…………………………………………………………………………………………..
Specific comments
Comment 1: Abstract.
By my opinion, in this section authors could state that the survey was carried out between
June-October 2018 in Badalona (Barcelona, Spain), involving patients aged 65 years and over. Also, the result regarding follow-up visits by the Functional Model healthcare team detected by nursing staff (MF/IM 7.19 (9.29)/4.2 (5.41), p:0.028) what is reported both in table
4 (p. 10, In-person consultations NUR BCU) and Results section (p. 9, line 259), where different
values are indicated, i.e., MF/MI 7.35 (9.50)/4.33 (5.47), p:0.028. Moreover, I would suggest to
include the abbreviation for home care (HC).
Response: The proposal to complete the information given in the abstract is appreciated. The presence of the data provided in the tables and in the explanatory text is noted. The acronym HC is used throughout the manuscript and is added, as suggested, in the abstract.
Comment 2: Introduction is comprehensive and clear, well done. Only I think that authors could include also in this section abbreviations for both traditional/integrated (IM) and functional (FM) models of home care (HC) (p. 2, line 83). In this respect, I would recommend also to check consistency of these terms throughout the whole paper (including the abstract), since in some parts conversely the abbreviations MI and MF are written.
Response: We agree with the suggested writing, we have corrected the error and reviewed it for correctness throughout the article.
Comment 3: Materials and Methods. This section is well detailed. However, I would ask authors why they use the future tense verb, given that the trial has already been carried out (e.g., This quasi-experimental ended in October 2 .).
Response: We agree that, considering the study is finished, verb tenses should be corrected. In the revised version of the manuscript, all tenses have been revised.
Comment 4: Also, in the Informed Consent Statement only caregivers will be asked to sign under the statement I would also suggest to provide a title for the first paragraph of Materials and Methods section (study design and ethical aspects?), and to add (at the end of the paragraph itself) the Trial registration information, that currently only in the abstract is reported (i.e., Registered in ClinicalTrials.gov, Identifier: NCT03461315; March 12,
2018).
Response: The reviewer's suggestions have been included in the text.
Some details follow:
Comment 5: Selection criteria. I would suggest to describe (e.g., questionnaire for the
assessment of cognitive impairment in older patients, scoring from 0 = maximum cognitive
impairment, to 10 = and to include the related reference (currently ref. n. 24 in this paper).
Response: The reviewer's suggestions have been included in the text. We understand that readers who are not familiar with these scales will find it more informative.
Comment 6: Patient recruitment and Study conduct. The indication that the study was carried out between June and October 2018 in both sections is reported. I would suggest to delete this information in one of them. Moreover, EuroQoL, IEXPAC, and Zarit could be described, and also the related reference could be put close to the respective test, e.g., EuroQoL, [17], IEXPAC [18] and Zarit [19].
Response: The reviewer's suggestions have been added to the text. We understand that readers who are not familiar with these scales will find it more informative. Following the suggestions, regarding the sections "Patient recruitment" and “Study conduct”, respectively, the text "was carried out between June and October 2018" and the duration of the study, “from June 2018 to October 2020” were added.
Comment 7: Endpoints and variables. Also in this section I think it could be better to put the related reference close to the respective test, e.g., Barthel scale Pfeiffer test Braden
test [25], social risk (TRIS) [26], and social status (Table 3) [27].
Response: The proposal is appreciated and was added to the text.
Comment 8: Statistical analysis. A sentence in Spanish language is reported (Se realizó un anàlisis intermedio al superar los 100 sujetos reclutados, con una potencia calculada para el tamaño muestral, 125 sujetos, del 35,5%). Moreover, in this sentence but they should be 126, as sum of 58 IM and 68 FM, and as stated in other parts of the paper (e.g., p. 7, lines 205 and 212).
Response: We are grateful for the correction.Both the translation and the number of patients (126) have been corrected in the text, since, as indicated, it is an error.
Comment 9: Results I would recommend authors to check the following aspects/discrepancies.
I suppose there is a mistake at p. 7, lines 203-204. Author All 183 patients potentially
eligible for the study were asked for consent, and 57 of them refused to participate in the study
(18 ABS gran sol and 39 from ABS Apenins (and this seems right), in Figure 1
authors indicate 18 patients who refused to participate in the study from ABS Apenins, and 39
from ABS gran sol. Moreover, authors could explain better what is t
related values are indicated only in Figure 1.
Response: We agree with reviewer #2's suggestion and have corrected the mistake.
Comment 10: p. 7, line 214: Table 2 is indicated, but it seems Table 3.
Response: Como en nuestro escrito no coincide la pagina ni la linea, entendemos que se refiere a la frase “Although the two Primary Care Centres were chosen due to the similar overall socio-demographic profile of the populations attended .As the page and the line do not match in our text, we understand that it is referring to the sentence "Although the two Primary Care Centres were chosen due to the similar overall socio-demographic profile of the populations attended (Table 2)", which reflects the sociodemographic data for which these two Primary Care Centers chosen for the study, as we consider them to be comparable.We added “(as shown in Table 2)” at the end of the sentence to avoid confusion for the reader.
Comment 11: p. 7, line 217: (CCP) (FM 80.9%, IM 89.8%), but in Table 3 it is IM 82,8.
Response: The correction is appreciated and modified in the text.
Comment 12: p. 7, line 219: ICIP but in Table 3 it is PIIC.
Response: The correction is appreciated and modified in the text.
Comment 13: p. 7, line 225: GMA , but in Table 3 it is AMG.
Response: The correction is appreciated and modified in the text.
Comment 14: p. 8, lines 234 and 236: what are IGV and VGI. I suppose they indicate the
same, but I cannot find an explanation of these abbreviations in the text.
Response: It is the same concept, we would like to apologize, it was a mistake when writing the acronym. We changed, throughout the text, the abbreviation to a much more appropriate acronym, Comprehensive Geriatric Assessment (CGA).
Comment 15: p. 8, lines 234-245: some statistics are reported but I cannot find them in tables (data not shown?). Moreover (lines 243-244), some values come from Table 4 (p. 9), first row, online referral to specialists (13(19.1%) MF, 4(6.9%) MI, p: 0.045). Other values (in person referral to specialists) seem attributable to Table 4 but are differently reported in the text, e.g.,
(0.65(0.89) MF, 0.79(1.2) MI, p: 0.088).
Response: These data are not included in any table and we have mentioned them in the text to avoid misunderstandings. Data errors in the text regarding visits to other specialists have been corrected.
Comment 16: p. 9, line 253: CT , but it is CAT in Table 4 (p. 9).
Response: The correction is appreciated and modified in the text.
Comment 17: p. 10, lines 281-284: these seem data not in table/not shown. This could be stated.
Response: These data are not included in any table and we have mentioned them in the text to avoid misunderstandings.
Tables
Comment 18: Table 3: the meaning of ATDOM could be included in the footnote.
Comment 19: Table 4, p. 10: In-person consultations NUR non- UBA I suppose it should be
non-BCU.
Comment 20: Table 4, p. 10 (from line 285): I think this should be Table 5. Moreover, in the footnote the meaning of the abbreviations RESPIR, A&E, PADES could be added. PADES could be included also within the table itself (last row).
Response Tables: We agree with the recommendations and suggested corrections. The recommendations have been added and the text has been corrected.
Comment 21: Discussion: this section is interestingly developed, but I would suggest authors to use short sentences in order to facilitate the reading. Some parts are a little difficult to follow (to me). Moreover, authors could distinguish better when data come from their own study or from
previous/other literature (for instance, p. 11, lines 292-303; lines 306-313).
Response: We agree that shorter sentences make it easier to understand the text. An attempt has been made to correct this. We have also tried to highlight the comments on the study to differentiate it from other studies to which it is compared to.
Comment 22: Limitations. By my opinion, it could be useful to explain (for the readers) what is Hawthorne effect (e.g., people have a different behavior when they know they are observed by researchers , and also add a reference. Moreover, (is it GMA, Adjusted Morbidity Groups?).
Response: The comment is appreciated and the explanation with the reference is added. The acronym was changed.
Conclusions are consistent with the evidence presented.
Comment 23: References are sufficiently updated. However, I would suggest to give a check, since some info seem lacking in some parts (e.g., publication date, number of pages, date of access of some links).
Response: Bibliography has been revised to correct the referencing.

Round 2
Reviewer 2 Report
Authors well addressed the suggested revisions. The paper seems clearer and with useful additional information. Some minor issues could however be solved, by my opinion, as follows.
Abstract (Material and methods): Authors write: “Two years follow-up of patients de 65 or older from two...” I suppose “de 65” is a typo, and probably it should be “of 65 years and older”.
In the “Informed Consent Statement” there is still a future tense verb to be corrected: “…only caregivers will be asked to sign under the statement..”. It should be “…only caregivers were asked to sign under the statement...”.
Paragraphs “Patient recruitment” and “Study conduct”. I suggested to delete the same info (i.e., the study was carried out between June and October 2018...) from one of them, since it‘s a duplication, but it is still in both sections.
Figure 1: I would suggest again to explain what is “lost of sample” (at least with a footnote in the figure).
Page 11, line 321: Authors write “Data not in table”, but then they delete this information, that by my opinion could conversely be useful for the readers.
References: some info seem still lacking in some parts (e.g.: ref. 2, p. 2276; ref. 8: 2007 and Journal of the American Medical Association). However, I suppose that such aspects will be further addressed in the final version of the manuscript.
Author Response
RESPONSES TO REVIEWER 2 COMMENTS.
We would like to thank reviewer2 for the review of the manuscript again. We have revised the manuscript by including the suggestions. We believe that, thanks to these suggestions, the manuscript is much clearer.
Comment 1*: Abstract (Material and methods): Authors write: “Two years follow-up of patients de 65 or older from two...+” I suppose “de 65” is a typo, and probably it should be “of 65 years and older”.
Response: The correction is appreciated and modified in the text
Comment 2*: In the “Informed Consent Statement” there is still a future tense verb to be corrected: “…only caregivers will be asked to sign under the statement..”. It should be “…only caregivers were asked to sign under the statement...”.
Response: We appreciate the comment and it has been modified in the text
Comment 3*: Paragraphs “Patient recruitment” and “Study conduct”. I suggested to delete the same info (i.e., the study was carried out between June and October 2018...) from one of them, since it`s a duplication, but it is still in both sections.
Response: The reviewer's suggestions have been included in the text.
Comment 4*: Figure 1: I would suggest again to explain what is “lost of sample” (at least with a footnote in the figure).
Response: The reviewer's suggestions have been included in the text.
: Comment 5*: Page 11, line 321: Authors write “Data not in table”, but then they delete this information, that by my opinion could conversely be useful for the readers.
Response: We especially appreciate this appreciation. We have been able to detect an error in the statistical calculation and we have been able to correct it.
Comment 6*: References: some info seem still lacking in some parts (e.g.: ref. 2, p. 2276; ref. 8: 2007 and Journal of the American Medical Association). However, I suppose that such aspects will be further
addressed in the final version of the manuscript.
Response: The reviewer's suggestions have been included in the text
